



# The Tall Tower Dataset. A unique initiative to boost wind energy research

Jaume Ramon[1], Llorenç Lledó[1], Núria Pérez-Zañón[1], Albert Soret[1], and Francisco J. Doblas-Reyes[1,2]

[1]Barcelona Supercomputing Center (BSC), c/ Jordi Girona, 29, Barcelona 08034, Spain
[2]ICREA, Pg. Lluís Companys 23, Barcelona 08010, Spain

**Correspondence:** Jaume Ramon (jaume.ramon@bsc.es)

**Abstract.** A dataset containing quality controlled wind observations from 222 tall towers has been created. Wind speed and wind direction measurements have been collected from existing tall towers around the world in an effort to boost the utilisation of these non-standard atmospheric datasets, specially within the wind energy and research fields. The observations taken at several heights greater than 10 metres above ground level have been retrieved from various sparse datasets and compiled in a unique collection with a common format, access, documentation and quality control. For the latter, a total of 18 Quality Control checks have been considered to ensure the high quality of the wind records. Non-quality-controlled temperature, relative humidity and barometric pressure data from the towers have also been obtained and included in the dataset. The Tall Tower Dataset (Ramon and Lledó, 2019a) is published in the repository EUDAT and made available at https://doi.org/10.23728/b2share.0d3a99db75df4238820ee548f35ee36b.

## 1  Introduction

Renewable energies have experienced the fastest growth among all electricity sources in the last few years (OECD/IEA, 2018, 2019). Together with solar photovoltaic, the wind power sector is leading this development, and the number of new wind farms and the installed capacity is currently facing an important increase worldwide (WindEurope, 2018; AWEA, 2019).

With higher shares of electricity generation depending on wind speed conditions, it is crucial to advance understanding of wind speed conditions at heights between 50 and 150 metres above ground —where current wind turbines are installed— and at multiple time-scales ranging from turbulence to mesoscale circulations, seasonal to decadal oscillations and climate change impacts. To characterise these features, high quality meteorological observations are needed.

Vast amounts of surface wind measurements taken at the standard height of 10 m above surface level do already exist and efforts have been made to compile the existing surface wind observations (Lott, 2004; Dunn et al., 2012; Klein Tank et al., 2002; Lucio-Eceiza et al., 2018a). However, meteorological data at turbine hub heights are much scarcer than surface observations. To take those measurements, a tall tower or met mast needs to be installed and instrumented. The basic structure of these masts consists of a high vertical tower reaching heights of 100 to 200 metres above ground with several platforms distributed along the vertical structure. It allows the placement of several wind sensors (i.e., anemometers and wind vanes) at different heights so that the vertical wind shear can be profiled. In addition, it is also typical to install several horizontal booms at each measuring



height oriented to different directions. Thus, more than one sensor per measurement level can be installed to correct or replace data from one of these redundant sensors in case it is affected by a technical failure or by the wind shadow produced by the mast itself. The physical structure of a tall tower as well as a typical instrumentation layout are illustrated in Figure 1.

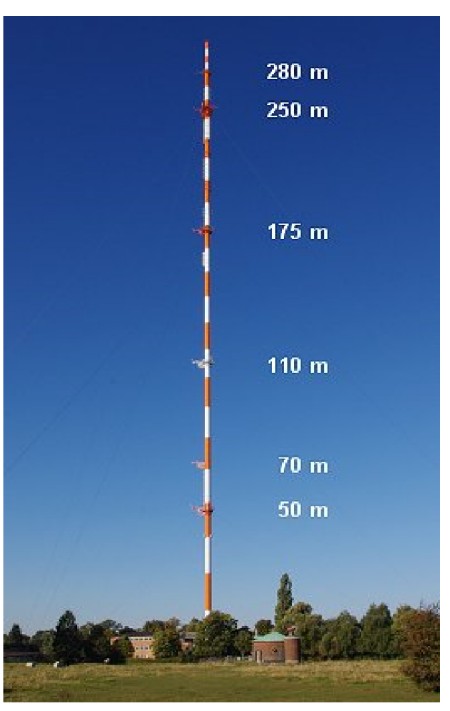
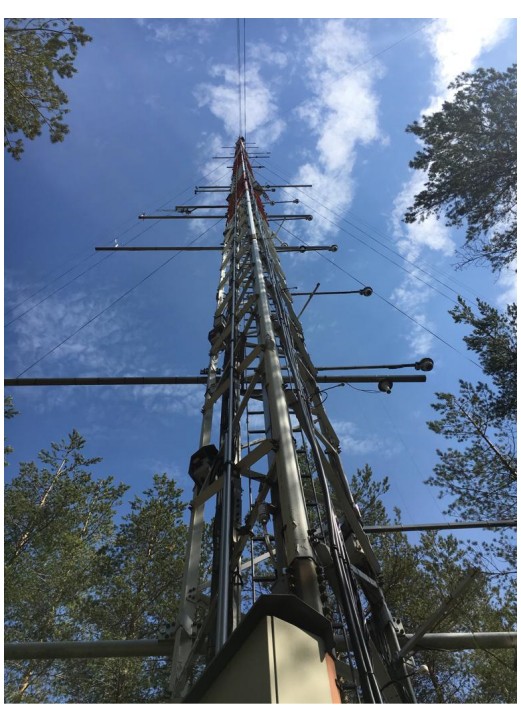

**Figure 1.** Left: Measuring levels at Hamburg university meteorological mast (Germany). *Source: https://icdc.cen.uni-hamburg.de.* Right: Arrangement of the instrumentation in booms at Hyytiälä forest met tower (Finland). *Courtesy of Jesús Yus-Díez.*

High-elevation observations are widely used in different initiatives to: a) evaluate of the wind resource characteristics and
derive wind power generation estimates (Brower et al., 2013); b) study local wind shear, turbulence and the dynamics of the Planetary Boundary Layer, PBL (Li et al., 2010); c) enhance or verify reanalysis products (Ramon et al., 2019; Decker et al., 2012); d) correct meteorological forecasts (Baker et al., 2003) and climate predictions (Torralba et al., 2017); or e) calibrate and verify wind atlas products (e.g., Troen and Petersen (1989); Fernando et al. (2018); Tammelin et al. (2013)).

Indeed, most of the existing met masts are owned by private companies belonging mainly to the wind energy industry. Wind
energy companies need to take those measurements prior to the construction of a new wind farm to characterise the wind speeds in the area and eventually ensure the return of the initial investment. Besides, some local effects such as topographic channelling, sea breezes, turbulence or vertical wind shear must be inferred because they can have a substantial impact on the electricity production (Hansen et al., 2012). Since the maintenance costs of these large and complex structures are rather expensive, the energy industry typically takes measurements for a relatively short period (1 or 2 years usually). Then the
towers are decommissioned, so the lack of long records of tall tower data reduces the possibilities to study, for example, wind





variability at seasonal to decadal time scales. In addition, private companies are usually reluctant to share the tall tower data with third parties, obstructing even more their further usages.

Fortunately, many of the initiatives from *a)* to *e)* also take tall tower measurements for their research and then the data are usually made freely accessible for non-commercial purposes. Derived from these diverse efforts devoted to boosting the utilisa-

tion of tall tower records, there exist various sparse datasets containing measurements from instrumented towers. Regrettably, they are often difficult to find or access. Furthermore, the lack of coordination in terms of formats, metadata, data access, and Quality Control (QC) hinder their usability outside the owner institution.

The INDECIS project (www.indecis.eu) is putting efforts to collect existing non-standard meteorological observations, among other aspects. In this paper, a dataset is presented, and the QC of the wind data is further detailed. The reader is

referred to Ramon and Lledó (2019b) to find complete information on the identification and collection of towers, data formatting and documentation. Sect. 2 of this article describes the main features of the dataset, as well as the data characteristics. The QC software suite is defined in Sect. 3. Then, a wrap up of the results after running the QC checks is presented in Sect. 4. The benchmark experiment carried out to test the robustness of the QC software is shown in Sect. 5. Finally, the conclusions are presented in Sect. 6.

## 2 Tall Tower Dataset description

The Tall Tower Dataset (Ramon and Lledó, 2019a) is a unique collection of data from 222 tall towers resulting from an exhaustive process of identification of existing masts and their later data retrieval. Figure 2 presents the global distribution of the sites, which is highly heterogeneous. Most of the masts are located in Asia (51%), mainly clustered in Iran resulting from a national campaign aimed to boost renewable energies at a country level. Then, tall towers appear more spatially distributed

over North America (23%) and Europe (16%), mirroring the important deployment of wind power that is taking place in those regions. Africa (8%), Oceania (1%) and Antarctica (1%) follow. Unfortunately, it has been hard to retrieve data from South America, so no records from this area can be found in the Tall Tower Dataset.

The height above the surface where the top sensor is located for each tower is also depicted in Figure 2. On the one hand, masts placed in historical observatories (i.e., often having more than 20 years of data) tend to be short, with heights ranging

between 18 and 50 metres above the ground and usually consist of one measuring level at the top of the pole. Two examples are the American masts in Barrow and Mauna Loa. On the other hand, modern towers often reach 100 to 200 metres of altitude. Indeed, most of the masts in northern Europe have been installed during the last 15-20 years and are generally taller than 80 m, usually reaching 150 to 200 m. However, the tallest structures are located in the USA reaching the exceptional height of 500 m, allowing the placement of sensors up there. The top anemometer at Walnut Grove tall tower in California is at 488 m above

ground level. The number of measuring levels in these masts is almost always higher than three, and up to eight in the case of the FINO met masts.

A list of the towers included in the Tall Tower Dataset, as well as their main characteristics such as the owner institution, country, geographic coordinates or explicit recording periods, can be found in Sect. S1 of the Supplementary Material. The



record lengths and other structural features such as height or instrumentation are quite diverse as they depend on the purpose they were designed for. Most of the towers are typically installed to provide in situ observations for experimental field campaigns within the research or industry fields. In this case, the tall towers are commonly referred to as meteorological masts or met masts, and they represent up to the 87% of all the tall towers in the dataset. However, other masts are installed over marine

platforms (11%) or at the top of lighthouses (1%) to monitor the coastal weather conditions. Finally, 1% of the towers are instrumented communication transmitters that take meteorological measurements at several platforms along with the antenna. Concerning the location, almost 80% of these tall towers are found inland while the other 20% are placed offshore.

Information indicating the representative features mentioned above is included in the dataset within the corresponding site metadata, which has been standardised for all the sites. This material was sometimes confusing, sparse or even missing in

the datasets distributed by the owner data centres, specially when it comes to the conventions in which the initial data were prepared. For example, if the time zone in which the time stamps were delivered was not specified, it could be challenging to discern whether they are provided in local time or using Universal Time Coordinated (UTC). Another example concerns the data units, which were not explicitly stated in a few cases either. In both of these confusing situations the data provider was contacted to confirm the original convention. Further information on the diverse standards in which the data were provided as

well as the final conventions employed in the Tall Tower Dataset can be found in Ramon and Lledó (2019b).

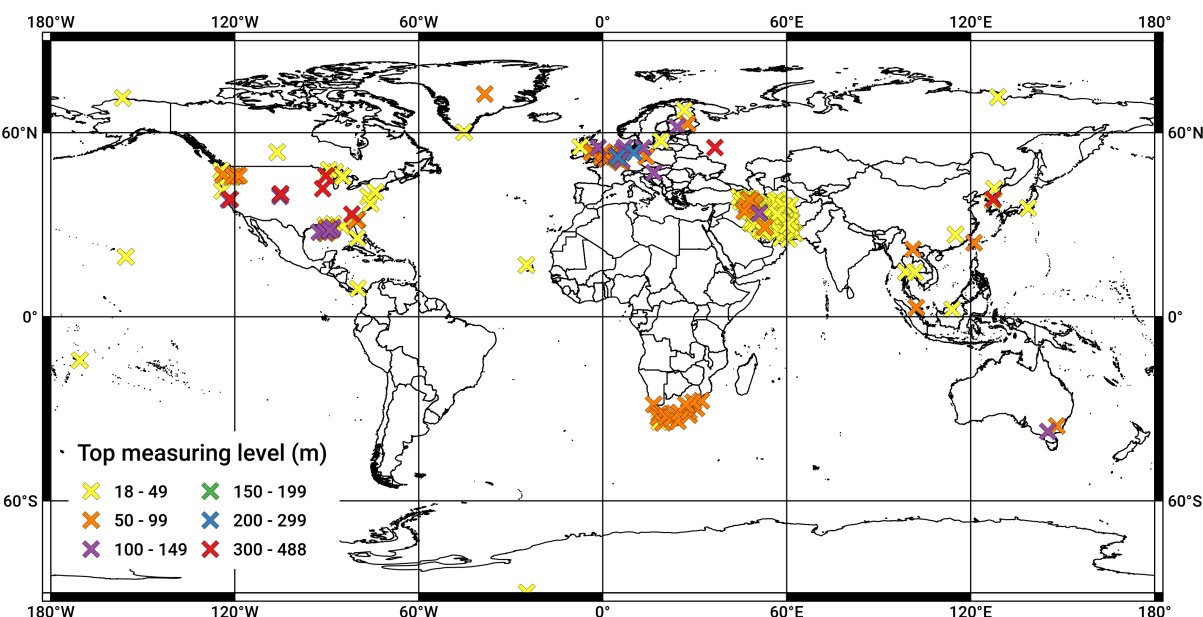

**Figure 2.** Global distribution of the 222 tall tower locations within the Tall Tower Dataset. Colours indicate the top measuring level for each tower.

The record length of the 222 time series is depicted in Figure 3(a) in terms of the reported time stamp sampling, which varies from 10-minutely to 1-hourly. Most of the series (i.e, a total of 172) provide 10-minutely averaged data, meeting the WMO





standard (WMO, 2007) for estimating mean wind speeds. The other 50 masts report 15-minute, 20-minute, 30-minute or hourly data. Indeed, it is worth noting that many of these towers have the longest series, spanning more than 30 years in some cases. Although some series reach up to 33 years of duration, 90% of the time series span less than 20 years. Nevertheless, several of these masts have been recently installed, and measurements are currently ongoing.

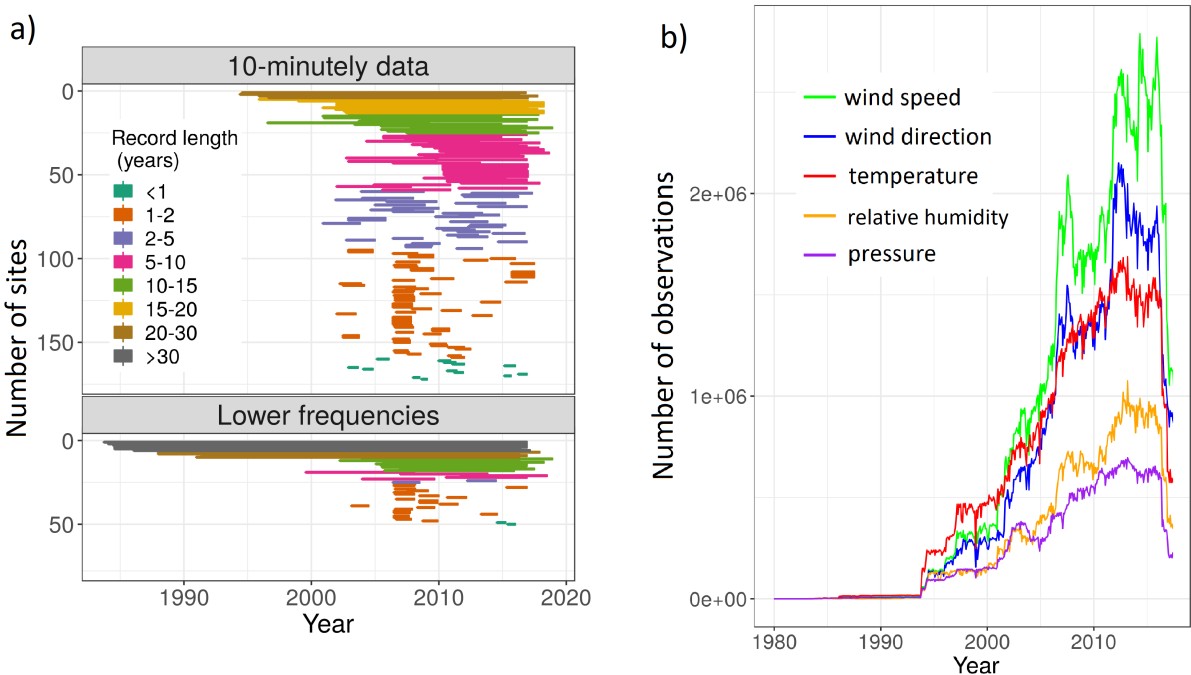

**Figure 3.** a) Time coverage of the 222 tall towers depending whether they report 10-minutely data (top) or lower resolutions up to 1-hourly data (bottom) and b) temporal evolution of the total amount of observations within the Tall Tower Dataset for wind speed (green), wind direction (blue), temperature (red), relative humidity (orange) and pressure (purple).

5     Concerning the data retrieving process, the primary efforts focused on collecting the largest amount of wind observations possible. Those records have been complemented with temperature, relative humidity and surface pressure data measured also at the different platforms along the tower. The temporal evolution of the amount of these five variables is plotted in Figure 3 (b). Most of the data falls within the $21^{st}$ century, with an important increase at the beginning of the millennium. Up to 2.7M of wind speed records have been retrieved for one single month, i.e. December 2015, which constitutes the month with the

10   maximum amount of wind speed data. In the case of wind direction, the month with the highest amount of records is October 2012 (2.1M of measurements). A decrease in the number of observations has been noticed from 2017 onwards. Generally, some of the data providers prefer to keep the most recent data and release them once measurements are preliminary checked for gross errors. Temperature, relative humidity and pressure are not always available. We note that the fewest records correspond to the barometric pressure, which is usually measured only at surface level (i.e., 2 m above ground level).



## 3   The Quality Control Software Suite for Tall Towers (QCSS4TT)

To ensure the high quality of tall tower wind data and guarantee the accuracy of any result derived from these records, a QC procedure needs to be carried out. The state-of-the-art has devoted efforts to QC wind data taken at surface stations (e.g., Dunn et al. (2012); Lucio-Eceiza et al. (2018b, c)). However, no QC software has been concretely designed to tackle the same

5    problem with tall tower observations, whose features vary considerably with respect to surface wind data (e.g., measurements are taken at higher altitudes, the spatial density of stations is considerably lower, etc.). Furthermore, specific aspects of the measuring techniques such as the parallel measurements at different platforms along the mast, or sensor redundancy at a given height can be taken into account to complement and enhance the typical QC.

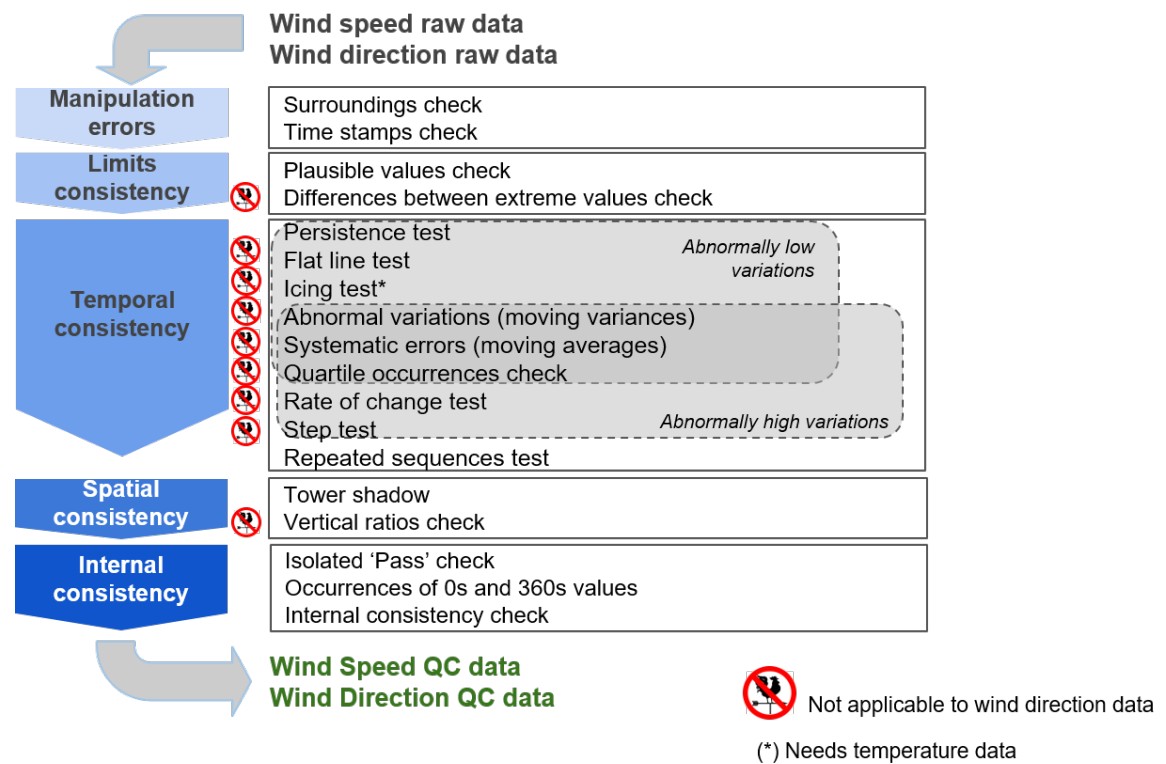

**Figure 4.** Summary of the workflow of the QCSS4TT routines applied over wind data within the Tall Tower Dataset.

After a review process of the existing QC routines, a set of 18 sequential QC tests (2 preliminary + 16 main tests) have

10    been selected and designed to be performed over wind measurements. The Quality Control Software Suite for Tall Towers (QCSS4TT) designed here is applied to all the wind speed and wind direction data within the Tall Tower Dataset, regardless of whether they were previously quality controlled or not by the providing institution. A general description of the QCSS4TT is presented below in this section. The software is fully described in Sect. S2 in the Supplementary Material.





The QC tests within the QCSS4TT ingest entire time series of winds at a specific height, whose time frequencies vary between one value every 10 minutes and one per hour. The recommended sequence for the application of the QC tests is presented in Figure 4. Checks are grouped in 5 categories depending on the purpose they were designed for. The two preliminary checks are designed to detect manipulation gross errors. Then, the 16 main QC tests ensure the limits, spatiotemporal and

5 internal consistency of the wind speed and wind direction time series. We note that the routines can be run independently, with the exception of the *Quartile occurrences* and *Isolated Pass* tests which feed on the output of other tests within the QCSS4TT. After deciding the appropriate order, the tests have been applied over the Tall Tower Dataset according to the flux diagram in Figure 4.

Table 1: Main QC tests summary. $ws$, $wd$ and $tmp$ stand for wind speed (in metres per second), wind direction (in meteorological degrees) and temperature (in Celsius degrees) respectively. The reader is referred to Sect. S2 in the Supplementary Material for detailed information on each of the tests.

| QC | Fail | Suspect | Remarks |
|---|---|---|---|
| Plausible values | $ws \notin [0, 113.2)$ $wd \notin [0, 360]$ | $ws \in (75, 113.2]$ | |
| Difference between extreme values of the distribution | | $max(ws) - max_{2nd}(ws) > max_{2nd}(ws)$ | Runs iteratively until the condition is not satisfied. |
| Persistence test | | if $A = \{x_t, ..., x_{t+60}\}$ is a set of 60 consecutive values, $max(A) - min(A) < 0.7$ for $ws$ and $max(A) - min(A) < 5$ for $wd$ | Skips calms ($ws \leq 0.5ms^{-1}$) |
| Flat line | 6 or more consecutive $ws$ values all equal, or 40 or more consecutive $wd$ values all equal. | 3 or more consecutive $ws$ values all equal, or 20 or more consecutive $wd$ values all equal | |
| Icing | $max(tmp) < 0$ and $max(ws) = 0$ for a 4-day period or longer. | | |



| QC | Fail | Suspect | Remarks |
|---|---|---|---|
| Abnormal variations | | $\sigma_i \notin [\overline{\sigma} \pm 4 * sd(\sigma)]$ where $\sigma$ is the distribution generated by the standard deviations of all the 30-day periods within a time series and $\sigma_i$ is the standard deviation of the $ith$ 30-day period | Only for $ws$. Disabled when more than 50% of data within a 30-day period are missing |
| Systematic errors | | $m_i \notin [\overline{m} \pm 4 * sd(m)]$ where $m$ is the distribution generated by the means of all the 30-day periods within a time series and $m_i$ is the mean of the $ith$ 30-day period | Only for $ws$. Disabled when more than 50% of data within a 30-day period are missing |
| Quartile occurrences | Given a period of time, all the observations fall above/below the first, second or third quartiles of the $ws$ distribution (see Table S2 of the Supplementary Material) | Given a period of time, all the observations fall above/below the first, second or third quartiles of the $ws$ distribution (see Table S2 of the Supplementary Material) | |
| Rate of change | $ws_{t+1} - ws_t \geq 3 * IQR$ | $2 * IQR \leq ws_{t+1} - ws_t < 3 * IQR$ | IQR is the interquantile range of the $ws$ distribution |
| Step test | $ws_{t+1} - ws_t \geq 20$ | | |





| QC | Fail | Suspect | Remarks |
|---|---|---|---|
| Repeated sequences | Repetition of a sequence of 20 or 30 consecutive $ws$ values, or repetition of a sequence of 30 consecutive $wd$ values. | | The values of the sequences need not be all equal. The maximum allowable length of the $ws$ sequence depends on the decimal places of the data |
| Tower shadow | | $ws$ values falling in the wake of the vertical structure | Only works if the tower contains redundant anemometers at the same level |
| Vertical ratios | Ratio between parallel $ws$ observations at different levels exceeds 30 units | Ratio between parallel $ws$ observations at different levels exceeds 15 units | Skips $ws$ lower than $1\ ms^{-1}$ |
| Isolated pass | Unflagged $ws$ or $wd$ values are surrounded by sequences of erroneous or missing data (see Table S3 of the Supplementary Material) | Unflagged $ws$ or $wd$ values are surrounded by sequences of suspect data (see Table S4 of the Supplementary Material) | This QC test needs to be run after the other routines. |
| Occurrences of 0s and 360s | The occurrence of 0s represents more than 30% of $ws$ values, or the occurrence of 0s and 360s represents more than 30% of $wd$ values | | Does not flag individual records but the entire time series |
| Internal consistency | $ws = 0$ and $wd \neq NA$[a] | | |

[a]NA: Not Available





**Table 2.** Flag levels definition

| Flag value | The observation... |
| --- | --- |
| 0 | has not been evaluated by three or more QC tests (partly QCed) |
| 1 | has passed all QC tests successfully |
| 2 | is deemed suspect |
| 4 | has failed at least one of the tests |
| 5 | is a calm wind |
| 9 | is missing |

The QCSS4TT starts with two preliminary tests. Firstly, the *Time stamps check* is carried out during the data formatting process and ensures that all the time stamps are included in the dataset and equally sampled according to the tower reporting frequency. Those time stamps that are either duplicated or wrongly spaced have been discarded, and those missing have been included, setting the corresponding record to 'Not Available' (NA). Secondly, the *Surroundings check* is performed by detecting

nearby elements that could potentially perturb the wind flow and then produce unreal records. To do so, detailed descriptions of the encircling area of the mast as well as their possible changes over time are required.

Then, the 16 main QC tests follow. A summary can be found in Table 1 and complete information can be encountered in Sect. S2 in the Supplementary Material. Most of them are standard checks typically performed over wind and other Essential Climate Variables such as temperature or precipitation. However, we propose here two new QC tests (the so-called *Tower*

*shadow* and *Vertical ratios* checks, respectively) to guarantee the spatial consistency of the data by considering the special characteristics of the tall tower measurements since classic inter-stational comparisons appear challenging due to poor spatial density of sites.

After running the QCSS4TT, a natural number (hereafter referred to as *QC flag* or *flag*, see Table 2) is attached to each observation according to its nature and/or level of confidence. To decide which flag should be assigned to each observation,

different threshold values have been set for each of the QC routines. The threshold selection has been based on the World Meteorological Organization (WMO) standards (WMO, 2007; Aguilar et al., 2003), QC software manuals (IOSS, 2017) or state-of-the-art bibliography (e.g., Jiménez et al. (2010)). Many of these standards, if not all, have been developed specifically for surface winds (i.e., 10 m winds), whose features vary importantly when compared with winds observed at higher altitudes, as in the current work. After a preliminary test of the thresholds over the wind data within the Tall Tower Dataset, it was

noted that some tests overestimated considerably the amount of erroneous data (also known as Type I errors, see Hubbard et al. (2004)). The WMO allows adjusting some of the fixed-value limits proposed in the WMO (2007) to reflect singular climate conditions more accurately. As the QCSS4TT aims to clean data from towers located all over the world regardless of the prevailing climate conditions in the area, thresholds need to be adjusted manually to not to deem wrong the general and particular climate features observed in the wide variety of world climates. It is also vital to take into account that this sensitive





experiment should reduce the number of Type I errors without increasing the number of invalid data that has been accepted by the tests (also referred to as Type II errors).

Based on these thresholds and the nature of the individual wind records, six different categories have been defined (Table 2), and each datum is flagged appropriately. The quality of a record is inferred automatically by checking if it passes all the tests successfully (flagged as '1'), passes the tests but might need further checks such as a visual inspection (hereafter referred to as *suspect* and marked as '2'), or fails at least one of the tests (flagged as '4'). When an observation is not considered suspect or wrong by any of the QC tests, additional levels may indicate that the observation was not evaluated by three or more tests (indicated as '0') or corresponds to a calm period ('5'). Finally, missing values are flagged uniformly (categorised as '9').

Wind records flagged as '4' are deemed to be erroneous data and thus, unreliable. They have been removed by changing the original record to NA. Suspect data remain unaltered as well as those observations that have not been evaluated by all the QC tests because they might be potentially correct and usable for some applications. But in case the user prefers to impose their own level of restriction, we also include the raw data jointly with the flag values resulting from the quality controlling. Therefore, the data user is able to filter the raw data based on the flag values. Still in those cases, we strongly discourage the usage of data marked as erroneous ('4').

## 4   Results of the application of the QCSS4TT

The QCSS4TT has been applied sequentially over the Tall Tower Dataset according to the flux diagram in Figure 4. We present here the global results obtained from the quality controlling of the Tall Tower Dataset, as well as a summary of the performance of the main tests.

As stated in Sect. 3, the *Surroundings check* needs detailed original metadata of the tower location. Unfortunately, this valuable information is not always available so the *Surroundings check* cannot be carried out over all the tower sites. The unique case when this QC test confirms that a series of wind speeds were disturbed by the surrounding forest occurs at Wallaby Creek met mast. After running the main QC routines, long sequences of wind speeds measured at the lowermost level of this met mast —placed at 10 m above surface— have been flagged as wrong. Then, a close look at the site metadata reveals that the canopy well exceeds the 10 metres height during all the recording period, thus reducing considerably the observed wind speeds. Hence, all the individual observations of the Wallaby Creek 10-metre wind series have to be used with caution, even those that have not been considered problematic by other tests.

Then, the main QC routines have examined each of the 240 371 908 wind speed and wind direction values individually in the Tall Tower Dataset and flagged them accordingly. After all this process, 228 780 679 values (95.2% of the total data) passed successfully all the checks and can be considered reliable. On the contrary, 6 827 880 observations (2.8%) have been considered erroneous by at least one of the QC tests. A 1.8% of the dataset is flagged as suspect. Some of the QC tests, particularly those that compute period-aggregated statistics such as moving averages or variances, require a minimum amount of data. Due to this constraint, a 0.2% of the data have not been evaluated by 3 or more QC tests to avoid the computation of such statistics with reduced sample sizes. Records identified as calms (i.e. wind speeds under $0.5~ms^{-1}$) have been also skipped on purpose

by a small group of tests, i.e. those that compute quotients between pairs of simultaneous observations. However, calms can be trusted as they passed successfully all the other QC checks. The percentage of calms is highly dependent on the geographical location of the tall tower. Met masts located in Southeast Asia contain the largest percentage of calms, reaching up to 24% of the total data.

The amount of data flagged by each test is considerably different as can be noticed in Figure 5, which depicts the percentage of data flagged as erroneous and suspect by the main QC tests. We note that both the *Flat line* and *Quartile occurrences* checks have flagged the largest amount data (1.74%). The former detected the most substantial amount of erroneous data (1.52%), followed by the *Repeated sequences* and *Quartile occurrences* tests (1.29% and 0.88%, respectively). Very few erroneous or suspect records have been detected by the *Vertical ratios* check, and the *Difference of extreme values* test has flagged no datum.

The *Occurrences of 0s and 360s values* test is not included in Figure 5 since this test does not flag individual records, but the entire time series according to their quality. Results for this QC show that no wrong or suspicious time series has been detected after the inspection of the frequency of appearance of null wind speeds and 0s and 360s wind direction values.

## 5   How reliable is the QCSS4TT?

The performance of the QCSS4TT needs to be assessed. Here, a benchmark experiment has been specifically designed to test

the ability of the QCSS4TT in detecting wrong values. In the following, the preparation of the experiment and their results are described.

The setup of the experiment consists of generating a set of presumably QC-free time series where a set of errors will be purposely introduced later on. The time series have been extracted at 50 randomly selected points from the ERA5 reanalysis (Copernicus Climate Change Service (C3S), 2017) global grid at hourly frequency, thus meeting the requirements of the

QCSS4TT concerning the time resolution. These time series span the 10-year period from 2007 to the end of 2016, which constitutes the time range with the largest amounts of records within the Tall Tower Dataset (Figure 3(b)). To better emulate the features of the tall tower data, we retrieve two parallel series at each of the 50 points. These wind speeds are those provided at 10 and 100 metres respectively.

The set of 50 series is replicated fourfold. Then, three of these series are firstly modified by introducing missing data at

random, either by erasing data individually or removing sequences of records. The percentages of missing data in these series are approximately 5%, 10%, and 20%, respectively. The introduction of missing records emulates the frequently observed sporadic sensor failures and no data periods within the wind speed series. Finally, one series is left with no datum set to missing.

The error 'seeding' process is carried out following the methodology in (Hubbard et al., 2004), where the performance of a

set of basic QC tests for temperature and precipitation data is assessed. In this publication, a subset of 2% of the total data is selected to be modified by introducing an error of magnitude:

$$E_{ix} = \sigma_x r_i \tag{1}$$



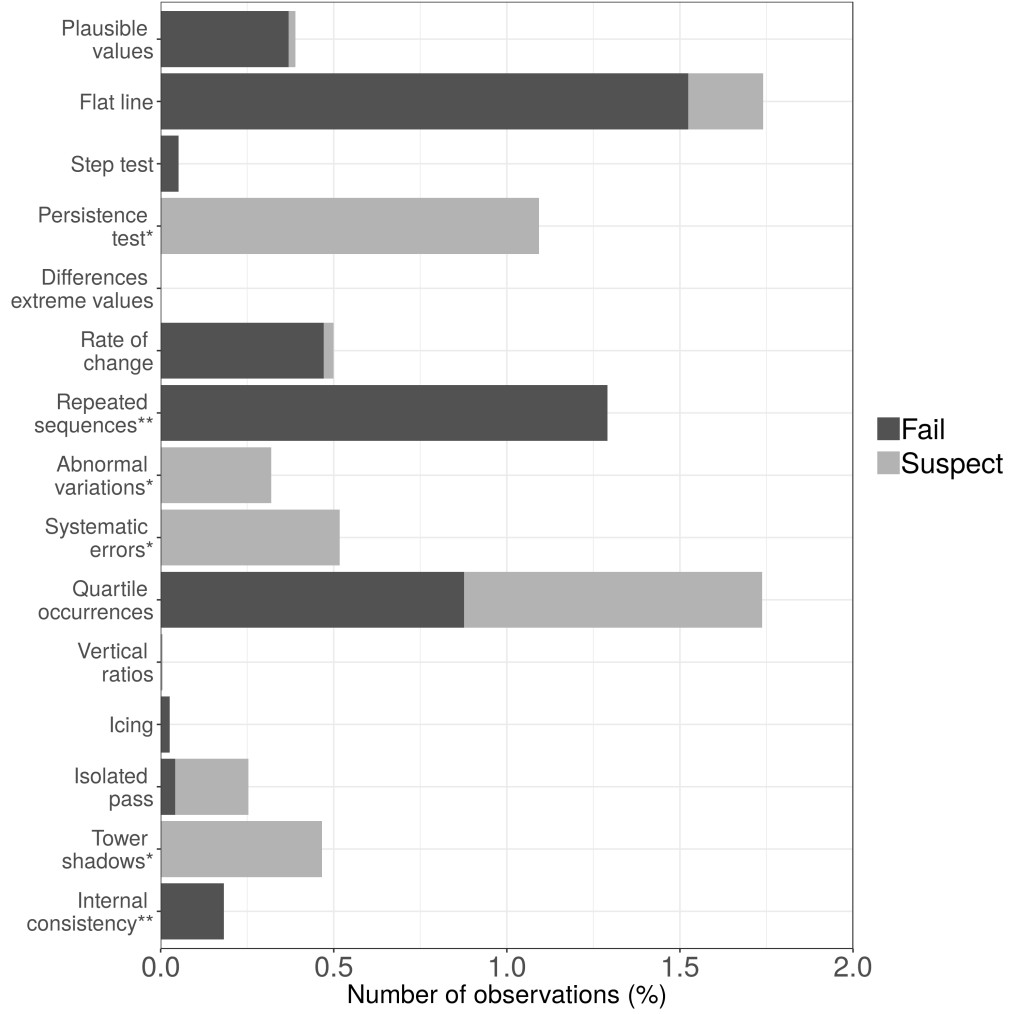

**Figure 5.** Percentage of data flagged as Fail (red) and Suspect (orange) by 15 of the main QC tests within the QCSS4TT. Asterisk (*) indicates that the QC test only flags data as suspect. Double asterisk (**) denotes that the QC test only flags data as erroneous.

where $\sigma_x$ is the standard deviation of the time series $x$ and $r_i$ is a randomly selected number generated using a uniform distribution ranging from -3.5 to +3.5 specifically for $ith$ observation. Once the errors are inserted, the QCSS4TT is executed.

Table 3 summarises the differences in the error detection depending on whether missing data are introduced or not into the wind series for both continental and oceanic locations. It is worth noting that the QCSS4TT shows a slight sensitivity to missing data, reducing the percentage of detected errors when the percentage of missing records increases. This decrease might be attributed to the fact that some QC tests are deactivated when a period with very sparse data is encountered. No important differences are noted between onshore and offshore sites, as the percentages of identified errors are quite similar. Results that





**Table 3.** Percentage of the detected errors (in %) depending on the proportion of data that were set to missing. The percentages of missing data are approximately 20%, 10%, 5%, and 0% (NA-free). The table differentiates between land and offshore locations.

|  | NA-free | 5% missing | 10% missing | 20% missing |
|---|---|---|---|---|
| Land sites | 40.3 | 40.2 | 39.8 | 39.1 |
| Offshore sites | 40.0 | 39.8 | 39.4 | 38.8 |
| Total | 40.2 | 40.0 | 39.6 | 39.0 |

follow in this section are presented for the set of series containing approximately 10% of missing data, which is close to the average of absent records within the Tall Tower Dataset (12.5%).

The QCSS4TT has detected on average nearly 40% of all the seeded errors (see Table 3). Indeed, this result is on the average of the percentage of detection observed for precipitation data in Hubbard et al. (2004), which was 30%-40% for complex terrain sites and 40%-50% for the other locations.

At this stage of the experiment, it is important to study the role of the random number $r$, and particularly its magnitude, which subsequently influences the size of the error $E$. Values of $r$ close to zero will introduce smaller errors, which will be less likely to be caught by any of the QC checks. Figure 6 presents the percentage of detection as a function of the $r$ values, which have been grouped in intervals of 0.5 units. We note that the QC tests detect most of the biggest errors. However, the percentage of detection decreases as the magnitude of $r$ does, as we expected. Thus, the smallest errors are usually skipped by all the QC tests. Indeed, this result mirrors the conservative philosophy employed in the threshold selection of the checks.

Finally, it has been observed that Type I errors have been made in 8% of the total data, corresponding mainly to suspect flagging.

## 6 Conclusions

Hub-height wind data are vital to assess the local wind flow features at heights ranging from 20 to 120 metres, where wind turbines are located. Nonetheless, the wind industry is not the only user of these observations, but also the research academy is interested in retrieving hub-height winds for their studies such as PBL experiments or the verification of climate products. Unfortunately, these non-standard climate data appear sparsely, and the lack of standardised formats, quality and metadata jeopardise their further usage. This is the first time when efforts were devoted to gather the most substantial possible amount of existing data measured at tall towers around the world, and perform an exhaustive QC assessment to eventually made them publicly available for non-commercial purposes in a standard format and access point. Wind speed, wind direction, temperature, pressure and relative humidity observations measured at different heights in 222 tall towers —owned mainly by public institutions such as universities, meteorological weather services or research centres— have been retrieved from sparse archives, compiled in a unique collection, quality controlled —in the case of wind speed and wind direction data—, and released under the name of the Tall Tower Dataset. Data from of 181 of these sites are stored in the EUDAT data repository and



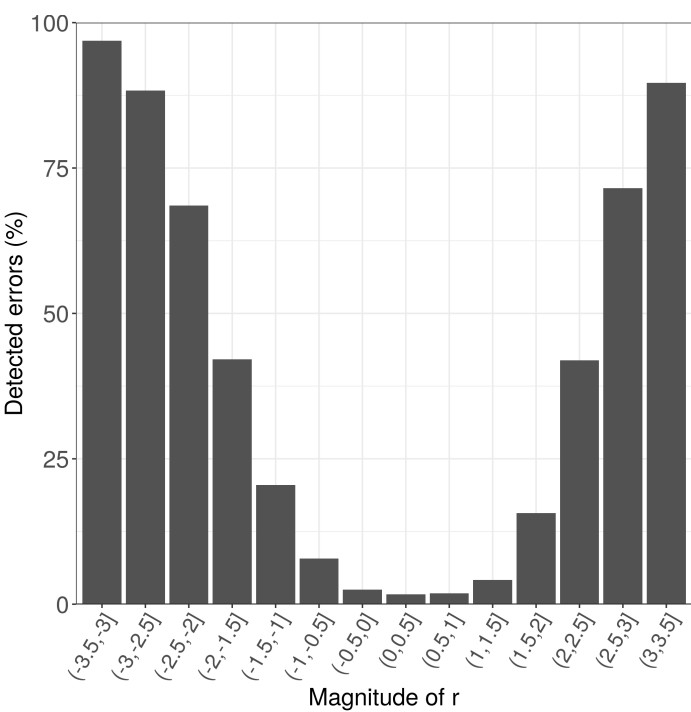

**Figure 6.** Percentage of detection of seeded errors as a function of the magnitude of the random number $r$

can be publicly accessed. Records from the other 41 towers are not available there since the authors of the Tall Tower Dataset do not own the observations and the data providers do not grant rights to share with third parties. Although some initiatives such as the Climate Data Store[1] are starting to appear to uniform and boost the free utilisation of climate observations, there is still some reluctance, mainly in Europe, concerning the contribution to open initiatives that derive in the inclusion of data in

5  public external archives, thus hindering their further usage.

To guarantee the reliability of the wind measurements, a QC software suite has been designed and applied over the Tall Tower Dataset, and the erroneous data have been removed. Some of the QC functions are coded to deal simultaneously with huge amounts of data so that the computation costs may be high specially when considering high resolution data. After the application of the QCSS4TT, the vast majority of the dataset (i.e, the 95.2% of the wind data) passed all the tests successfully.

10  A benchmark experiment based on Hubbard et al. (2004) has been designed to assess the efficacy of the QCSS4TT in detecting wrong wind speed data. The exercise is based on the detection of a set of seeded errors introduced in 100 wind time hourly series at 50 randomly selected locations obtained using the ERA5 reanalysis. On average, the 40% of these seeded errors have been identified, even though the magnitude of the error is sometimes close to zero and therefore, difficult to detect. This result agrees with the obtained by the previously mentioned publication, thus assuring the reliability of the QCSS4TT results.

[1]https://cds.climate.copernicus.eu/
We do not perform any analogue experiment for wind direction data since the nature of these data requires a more complex exercise.

Even though some tall towers have been decommissioned recently due to several different reasons, most of the locations within the Tall Tower Dataset continue taking measurements that could be added to the collection in a near future. Besides, the
authors of this work are open to receive useful inputs on new tower locations not included in the Sect. S1 of the Supplementary Material, and whose data could be potentially added to the Tall Tower Dataset in future updates. Enlarging the collection of these non-standard climate data and increasing the density of stations may allow, for instance, further quality checks by means of inter-station comparisons with nearby tall towers.

*Code and data availability.* Records from 181 out of the 222 tall towers within the Tall Tower Dataset (Ramon and Lledó, 2019a) are
publicly accessible through the following EUDAT repository: https://doi.org/10.23728/b2share.0d3a99db75df4238820ee548f35ee36b. The QCSS4TT code as well as an explanatory vignette is also available via GitLab at: https://earth.bsc.es/gitlab/jramon/INDECIS-QCSS4TT.

*Author contributions.* JR retrieved and formatted the tall tower data, produced the QCSS4TT code and carried out the benchmark experiment. He also wrote the first draft of this manuscript. The work has been done under the supervision of LL, who conceived the research, gave advice on the data collection and assisted JR in several IT issues and code debugging. NPZ came up with the design of the benchmark experiment
as well as some of the graphics to visualise the results. AS and FJDB also supervised this work and facilitated the data approaching and retrieving. All authors contributed to the analysis of the results and to the writing and editing of the paper.

*Competing interests.* The authors declare that they have no conflict of interest.

*Acknowledgements.* The authors acknowledge the funding support from INDECIS co-funded by the H2020 ERA-net ERA4CS (GA 690462) and the MICINN grant BES-2017-082216 ("Ayudas para contratos predoctorales"). Very special thanks are given to Enric Aguilar for his
useful advice on the QC production. Authors are grateful to Pierre-Antoine Bretonnière, Nico Manubens and Javier Vegas for their technical support at different stages of this work. We also acknowledge the ECMWF as the ERA5 reanalysis production centre. Finally, authours would like to acknowledge explicitly all the principal investigators and owner institutions of the 222 tall towers within the Tall Tower Dataset for sharing their valuable climate data. We thank the National Data Buoy Center (NDBC), the Earth System Research Laboratory (ESRL) and the Physical Monitoring Program of the Smithsonian Tropical Research Institute (STRI). Special thanks are given to all the contributors to the
national mast database by the Renewable Energy and Energy Efficiency Organization in Iran. Berms Aspen data was provided by NCAR/EOL under the sponsorship of the National Science Foundation. We are grateful to Dr. Jinkyu Hong and Dr. Young-San Park (Korea Meteorological Administration) for facilitating Boseong mast data retrieval. Authors would like to thank the Bonneville Power Administration (BPA), the CESAR observatory and Dr. Fred Bosveld (KNMI) for providing Cabauw data, the South Africa Weather Service (SAWS), the Wind Atlas for South Africa project (WASA), the United Kingdom Met Office (UKMO). Many thanks to Lena Kozlova (University of Exeter) for sharing





Cape Verde tall tower data. Acknowledgements to The Crown State, the BMWi (Bundesministerium fuer Wirtschaft und Energie, Federal Ministry for Economic Affairs and Energy ) and the PTJ (Projekttraeger Juelich, project executing organization), the data contributors to the AsiaFlux database, Dr. Ingo Lange (Hamburg University), Dr. Laszlo Haszpra (Hungarian Met Service), Dr. Jan Schween (University of Cologne) and Dr. Frank Beyrich (Deutscher Wetterdienst) for facilitating the approach to Hamburg University, Hegyhatsal, Juelich and

5    Lindenberg masts data, respectively. We also aknowledge Dr. Rolf Neubert (University of Groningen), Met Éireann, the NREL National Wind Technology Center (NWTC, Jager and Andreas (1996)), Dr. Anna Rutgersson (Uppsala University), Prof. Gil Bohrer (The Ohio State University), and the NoordzeeWind B.V. (NZWBV) and/or its (sub)contractors. Thanks are also given to Ole Ziemer (Nukissiorfiit Hovedkontoret) for providing the data and Kurt S. Hansen (DTU) for putting us in contact with the data responsible. Park Falls tower data (Davis et al., 2003) was provided by AmeriFlux, which funding was provided by the U.S. Department of Energy's Office of Science. Thanks

10   are given to the FLUXNET community who provided valuable data from different masts and also to Christy Schultz (GMD Met - NOAA) for approaching us to South Pole mast data. Credit is also given to the original sources of the Tumbarumba met mast data, and the Vielsalm data manager Anne de Ligne and data provider Tanguy Manise.



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
