# Peer review of "The Tall Tower Dataset. A unique initiative to boost wind energy research"

_Earth System Science Data, 2019_

## Referee Comment (RC1) · Anonymous Referee #1 · 25 Oct 2019

General comments The proposed manuscript is considered fy this referee as an example of a paper that clearly explains the steps followed to compile a new dataset of wind data and, mainly, the quality control process applied. The authors have a special interest in explaining in detail the QC process, as well as its reliability. The documentation of the dataset and the quality control software is accurate and are another strong point. My recommendation is to accept the manuscript with some minor revisions.

2) Tall tower dataset description. In the text (or even in the supplementary material) there is no information about how the wind direction and speed data is treated, i.e. scalar or vector averaging. This could be interesting as it has and impact in those cases with greater wind direction variance. Some information could be added, even if no information is available from the provider.

[Figure]

In the same point, information on the total temporal coverage of the dataset is not supplied (i.e., extreme years).

3) The Quality Control Software Suite for Tall Towers (QCSS4TT) As the QC runs on data at a specific height, somewhere in the text it should appear that it is an absolute quality control, just to clarify which kind of QC process is being applied.

4) Results of the application of the QCSS4TT Although the presence of missing data is cited in the text, no reference is made to the percentage of the gaps with respect to the total number of observations. It could be interesting to know this information, before and after the application of the method. This information could be added in figure 5 (just as a suggestion).

Congratulations for the manuscript.

---

## Referee Comment (RC2) · Anonymous Referee #2 · 28 Oct 2019

**1   General comments**

The manuscript presents the wind data of 222 tall towers, of which 181 are made publicly available. This is a very valuable dataset, which is well described, with a large emphasis on data quality control.

**2   minor remarks**

1. Please add the height of the tower, the number of measuring levels, and the time stamp sampling (10min, hourly, gusts) to table S1 (switch the table to landscape).

[Figure]

2. Please also add these items to the 0-INDEX_public.csv file

3. The end date of S1 is not always correct; e.g. Cabauw (NL) is said to have ended in 2017, whereas it is still operational. It should be advantous if it is indicated which towers are still operational.

4. some datasets (e.g. Cabauw (NL)) are not in the dataset, although the data is freely downloadable (after login). Maybe a link can be given to the place where these data can be downloaded?

5. Page 5, line 12. Maybe the increased use of lidars in stead of met masts is another reason for the decrease after 2017?

6. At least at a part of the towers maximum gusts will be measured as well. Addition of these gust observations to the dataset would be of great value.

---

## Author Comment (AC1) · 25 Nov 2019

Authors want to thank the reviewer for his/her comments, suggestions and thoughts on the paper. Responses to the reviewer's comments are described in detail below in blue colour. These reviews will be addressed accordingly when submitting the revised paper.

RC.1) The proposed manuscript is considered by this referee as an example of a paper that clearly explains the steps followed to compile a new dataset of wind data and, mainly, the quality control process applied. The authors have a special interest in explaining in detail the QC process, as well as its reliability. The documentation of the dataset and the quality control software is accurate and are another strong point. My

recommendation is to accept the manuscript with some minor revisions.

AC.1) Authors would like to thank the referee for her/his positive comments on this work.

RC.2) Tall tower dataset description. In the text (or even in the supplementary material) there is no information about how the wind direction and speed data is treated, i.e. scalar or vector averaging. This could be interesting as it has an impact in those cases with greater wind direction variance. Some information could be added, even if no information is available from the provider.

AC.2) Temporal resolutions of the published data are those in which records have been provided. As the reviewer points out, information concerning the treatment of the raw data by the provider is scarce. However, and after an exhaustive review process of all towers' metadata, we found out that 4 of the towers in Table S1 derive averaged values by averaging the wind components, whereas 2 take averages over the instantaneous modules and direction. The remaining 216 of the towers' metadata does not provide any information on how wind speed and direction are treated.

It may be worth reviewing best practices, such as those provided by the World Meteorological Organisation, on how instantaneous records (usually high-frequency data) must be averaged into minutely/hourly values. Also, the impact the reviewer mentions, as well as the effects on the uncertainty of the averaged values, can also be discussed in the revised version of the manuscript.

RC.3 ) In the same point, information on the total temporal coverage of the dataset is not supplied (i.e., extreme years).

AC.3) This information will be added to the text.

RC.4) The Quality Control Software Suite for Tall Towers (QCSS4TT). As the QC runs on data at a specific height, somewhere in the text it should appear that it is an absolute quality control, just to clarify which kind of QC process is being applied.

AC.4) Most of the QC tests (15 out of the 16 main QC tests) run on data at a specific height. The remainder test, i.e. the Vertical Ratios check, compares data measured at different tower levels, so it cannot be classified as an absolute test. However, all of the QC tests are intra-stational tests, which we think could be mentioned in the revised text for clarification.

RC.5) Results of the application of the QCSS4TT. Although the presence of missing data is cited in the text, no reference is made to the percentage of the gaps with respect to the total number of observations. It could be interesting to know this information, before and after the application of the method. This information could be added in figure 5 (just as a suggestion).

AC.5) We will compute this information which we also think is of interest. Thank you for suggesting an appropriate place to add it.

RC.6) Congratulations for the manuscript.

AC.6) Thank you very much for the positive appreciation of this manuscript.

---

## Author Comment (AC2) · 25 Nov 2019

AC.0) Authors want to thank the reviewer for his/her comments, suggestions and thoughts on the paper. Responses to the reviewer's comments are described in detail below in blue colour. These reviews will be addressed accordingly when submitting the revised paper.

RC.1) The manuscript presents the wind data of 222 tall towers, of which 181 are made publicly available. This is a very valuable dataset, which is well described, with a large emphasis on data quality control.

AC.1) Authors would like to acknowledge the positive appreciation of this work.

RC.2) Please add the height of the tower, the number of measuring levels, and the

timestamp sampling (10min, hourly, gusts) to table S1 (switch the table to landscape)

AC.2) This information will be added to table S1.

RC.3) Please also add these items to the 0-INDEX_public.csv file.

AC.3) 0-INDEX_public.csv file will be modified accordingly.

RC.4) The end date of S1 is not always correct; e.g. Cabauw (NL) is said to have ended in 2017, whereas it is still operational. It should be advantageous if it is indicated which towers are still operational.

AC.4) Thank you for noting this. We agree that adding this information is useful. The information on which towers are operational –according to tower station website, meta-data, and data provider– will be added to table S1.

RC.5) some datasets (e.g. Cabauw (NL)) are not in the dataset, although the data is freely downloadable (after login). Maybe a link can be given to the place where these data can be downloaded?

AC.5) This information can be encountered within the NetCDF files in which the data is presented. A global attribute ('links') indicates the tower station reference website.

Besides this, we are currently preparing a website to disseminate this dataset further, which is not finished yet. Links to original tower databases will also be provided there. We aim to include the location of this website in the revised version of our manuscript.

RC.6) Page 5, line 12. Maybe the increased use of lidars instead of met masts is another reason for the decrease after 2017?

AC.6) This is indeed a very interesting observation. However, we believe that the sudden decrease in the number of data is unlikely to have been produced by a sudden increase in the number of lidars. We observe the decrease in data in all the stations at a specific date. Even supposing that tower responsible were to replace tall towers by lidars after a specific date, they may have to maintain tall tower measurements parallelly for a while as a reference for assessing the difference between lidar data (e.g., possible biases).

During the data retrieval conducted during 2017 and the first half of 2018, we contacted several data providers. Some of them mentioned that the most recent data are not usually released because they prefer to perform first some QC tests to detect possible gross errors, so that we believe this is the main reason that explains the decay in the number of data after 2017.

On the other hand, we realised that no information on the increase of lidars to measure atmospheric profiles had been mentioned in the introduction section. We will add some notes on this.

RC.7) At least at a part of the towers maximum gusts will be measured as well. Addition of these gust observations to the dataset would be of great value.

AC.7) We agree that information on gusts is valuable for many studies. However, only 22% of the towers provide this information along with the wind averaged values. We will consider including this information in the future. QC checks have not been prepared for gusts. This would need specific methods that have not been investigated here yet.

––––––––––––––––––––––––––––

---

## Author Response (AR1)

Jaume Ramon
Barcelona Supercomputing Center
Jordi Girona 31
Barcelona, Spain 08028
jaume.ramon@bsc.es

**Kirsten Elger**
GFZ-Postdam, DE
Handling Topical Editor
*Earth System Science Data*

20 December, 2019

Dear Dr. Elger,

We have carefully reviewed the comments posted on Earth System Science Data Discussions and revised the paper accordingly. Our responses are given below in a point-by-point manner. Changes in the manuscript are presented in blue colour and can be found in the present document. In this revision, we have also uploaded a new version of the dataset in EUDAT. A new DOI has been obtained, although it has not been activated yet. We expect it to be active in the next few days. In order to avoid delaying this re-submission, you can access the revised dataset here: https://b2share.eudat.eu/records/136ecdeee31a45a7906a773095656ddb until the DOI is active. In this version of the dataset, only one file (i.e., 0-INDEX_public.csv) has been updated according to comment M.2 of Referee #2.

We hope the revised version is now suitable for publication and look forward to hearing from you in due course.

Sincerely,

Jaume Ramon
PhD student, Earth Sciences department
Barcelona Supercomputing Center

Authors want to thank the two reviewers for their comments, suggestions and thoughts on the paper. Responses to the reviewers' comments are described in detail below in blue colour. These reviews have been addressed accordingly in the submitted the revised paper. Pages and line numbers included in some of the below comments refer to the marked-up manuscript file.

**Referee #1:**

**G.1)** The proposed manuscript is considered by this referee as an example of a paper that clearly explains the steps followed to compile a new dataset of wind data and, mainly, the quality control process applied. The authors have a special interest in explaining in detail the QC process, as well as its reliability. The documentation of the dataset and the quality control software is accurate and are another strong point. My recommendation is to accept the manuscript with some minor revisions.

Authors would like to thank the referee for her/his positive comments on this work.

**G.2)** Tall tower dataset description. In the text (or even in the supplementary material) there is no information about how the wind direction and speed data is treated, i.e. scalar or vector averaging. This could be interesting as it has an impact in those cases with greater wind direction variance. Some information could be added, even if no information is available from the provider.

Temporal resolutions of the published data are those in which records have been provided. As the reviewer points out, information concerning the treatment of the raw data by the provider is scarce. However, and after an exhaustive review process of all towers' metadata, we found out that 4 of the towers in Table S1 derive averaged values by averaging the wind components, whereas 2 take averages over the instantaneous modules and direction. The remaining 216 of the towers' metadata does not provide any information on how wind speed and direction are treated.

We have reviewed best practices, such as those provided by the World Meteorological Organisation, on how instantaneous records (usually high-frequency data) must be averaged into minutely/hourly values. Also, the impact the reviewer mentions, as well as the effects on the uncertainty of the averaged values, is also discussed in the revised version of the manuscript (see page 4, lines 26-30).

In the same point, information on the total temporal coverage of the dataset is not supplied (i.e., extreme years).

This information has been added to the text (see Abstract, line 2, and page 4, line 32).

**G.3)** The Quality Control Software Suite for Tall Towers (QCSS4TT). As the QC runs on data at a specific height, somewhere in the text it should appear that it is an absolute quality control, just to clarify which kind of QC process is being applied.

Most of the QC tests (15 out of the 16 main QC tests) run on data at a specific height. The remainder test, i.e. the Vertical Ratios check, compares data measured at different tower levels, so it cannot be classified as an absolute test. However, all of the QC tests are intra-stational tests, which has been mentioned in the revised text for clarification (see page 6, line 12).

**G.4)** Results of the application of the QCSS4TT. Although the presence of missing data is cited in the text, no reference is made to the percentage of the gaps with respect to the total number of observations. It could be interesting to know this information, before and after the application of the method. This information could be added in figure 5 (just as a suggestion).

We have computed this information which we also think is of interest. The initial percentage of missing data in the raw dataset was 12.1% (page 5, lines 1-2), whereas the percentage is increased to 14.6% (see page 12, lines 16-17).

Congratulations for the manuscript.

Thank you very much for the positive appreciation of this manuscript.

**Referee #2:**

**G.1)** The manuscript presents the wind data of 222 tall towers, of which 181 are made publicly available. This is a very valuable dataset, which is well described, with a large emphasis on data quality control.

Authors would like to acknowledge the positive appreciation of this work.

**M.1)** Please add the height of the tower, the number of measuring levels, and the timestamp sampling (10min, hourly, gusts) to table S1 (switch the table to landscape)

This information has been added to Table S1.

**M.2)** Please also add these items to the 0-INDEX_public.csv file.

0-INDEX_public.csv file has been modified accordingly. Please, note that changes on the 0-INDEX_public.csv file imply that a new version of the Tall Tower Dataset has been created and uploaded to the EUDAT repository. The link to this new version has been included in the revised manuscript.

**M.3)** The end date of S1 is not always correct; e.g. Cabauw (NL) is said to have ended in 2017, whereas it is still operational. It should be advantageous if it is indicated which towers are still operational.

Thank you for noting this. We agree that adding this information is useful. The information on which towers are operational --according to tower station website, metadata, and data provider-- has been added to table S1.

**M.4)** some datasets (e.g. Cabauw (NL)) are not in the dataset, although the data is freely downloadable (after login). Maybe a link can be given to the place where these data can be downloaded?

This information can be encountered within the NetCDF files in which the data is presented. A global attribute ('links') indicates the tower station reference website.

Besides this, we have prepared a website to disseminate this dataset further (see http://talltowers.bsc.es). Links to original tower databases are also provided there. The location of this website has been included in the revised manuscript, within the 'Code and data availability section'.

**M.5)** Page 5, line 12. Maybe the increased use of lidars instead of met masts is another reason for the decrease after 2017?

This is indeed a very interesting observation. However, we believe that the sudden decrease in the number of data is unlikely to have been produced by a sudden increase in the number of lidars. We observe the decrease in data in all the stations at a specific date. Even supposing that tower responsible were to replace tall towers by lidars after a specific date, they may have to maintain tall tower measurements parallelly for a while as a reference for assessing the difference between lidar data (e.g., possible biases).

During the data retrieval conducted during 2017 and the first half of 2018, we contacted several data providers. Some of them mentioned that the most recent data are not usually released because they prefer to perform first some QC tests to detect possible gross errors, so that we believe this is the main reason that explains the decay in the number of data after 2017.

On the other hand, we realised that no information on the increase of lidars to measure atmospheric profiles had been mentioned in the introduction section. We have added some notes on this (see page 2, lines 5-9).

**M.6)** At least at a part of the towers maximum gusts will be measured as well. Addition of these gust observations to the dataset would be of great value.

We agree that information on gusts is valuable for many studies. However only 22% of the towers provide this information along with the wind averaged values. We will consider including this information in the future. QC checks have not been prepared for gusts. This would need specific methods that have not been investigated here yet.

[revised manuscript text omitted]

---

## Author Response (AR2)

Jaume Ramon
Barcelona Supercomputing Center
Jordi Girona 31
Barcelona, Spain 08028
jaume.ramon@bsc.es

**Kirsten Elger**
GFZ-Postdam, DE
Handling Topical Editor
*Earth System Science Data*

20 January, 2020

Dear Dr. Elger,

We have carefully addressed your request sent on 16th January, and we have included the note (in blue colour in the marked-up manuscript) on the links of those datasets not included in the Tall Tower Dataset. Please, refer to lines 27-32 on page 16 of the manuscript.

We hope the revised version is now suitable for publication and look forward to hearing from you in due course.

Sincerely,

Jaume Ramon
PhD student, Earth Sciences department
Barcelona Supercomputing Center

[revised manuscript text omitted]